# *Euphausia pacifica* (North Pacific Krill): Review of Chemical Features and Potential Benefits of 8-HEPE against Metabolic Syndrome, Dyslipidemia, NAFLD, and Atherosclerosis

**DOI:** 10.3390/nu13113765

**Published:** 2021-10-25

**Authors:** Nanae Ishida, Hidetoshi Yamada, Masamichi Hirose

**Affiliations:** 1Department of Pathophysiology and Pharmacology, School of Pharmaceutical Sciences, Iwate Medical University, Iwate 028-3694, Japan; nishida@iwate-med.ac.jp; 2Faculty of Life & Environmental Science, Teikyo University of Science, Tokyo 120-0045, Japan; hyamada@ntu.ac.jp

**Keywords:** *Euphausia pacifica*, 8-HEPE, N-3 polyunsaturated fatty acids, nonalcoholic fatty liver disease

## Abstract

Marine n-3 fatty acids are well known to have health benefits. Recently, krill oil, which contains phospholipids, has been in the spotlight as an n-3 PUFA-containing oil. *Euphausia pacifica* (*E. pacifica*), also called North Pacific krill, is a small, red crustacean similar to shrimp that flourishes in the North Pacific Ocean. *E. pacifica* oil contains 8-hydroxyeicosapentaenoic acid (8-HEPE) at a level more than 10 times higher than *Euphausia superba* oil. 8-HEPE can activate the transcription of peroxisome proliferator-activated receptor alpha (PPARα), PPARγ, and PPARδ to levels 10, 5, and 3 times greater than eicosapentaenoic acid, respectively. 8-HEPE has beneficial effects against metabolic syndrome (reduction in body weight gain, visceral fat area, amount of gonadal white adipose tissue, and gonadal adipocyte cell size), dyslipidemia (reduction in serum triacylglycerol and low-density lipoprotein cholesterol and induction of serum high-density lipoprotein cholesterol), atherosclerosis, and nonalcoholic fatty liver disease (reduction in triglyceride accumulation and hepatic steatosis in the liver) in mice. Further studies should focus on the beneficial effects of North Pacific krill oil products and 8-HEPE on human health.

## 1. Introduction

It is well known that the liver has detoxification, metabolic, and storage functions. Consequently, when the liver is injured by disease, these functions are impaired. Obesity-related nonalcoholic fatty liver disease (NAFLD), which is a hepatic steatosis arising in the absence of alcohol intake, impairs the liver’s functions. NAFLD is known to participate in the development of other liver diseases, such as liver cirrhosis and hepatocellular carcinoma [1,2,3]. NAFLD is also associated with the development of diseases other than the liver, such as cardiovascular disease [4,5] and cognitive impairment [6,7,8]. These results indicate that the treatment of NAFLD is important to prevent several diseases and maintain normal liver function. Moreover, because metabolic syndrome (central adiposity, hyperglycemia, dyslipidemia, and arterial hypertension), weight gain, and insulin resistance/diabetes are risk factors for NAFLD [9,10], the elimination of these factors is important for suppressing the appearance of NAFLD, which would prevent the NAFLD-related diseases described above.

Marine n-3 polyunsaturated fatty acids (PUFA) are well known to have health benefits [11]. A diet containing n-3 PUFA, including eicosapentaenoic acid (EPA) and docosahexaenoic acid (DHA), reduces the incidence and mortality of cardiovascular disease through multiple mechanisms, such as the reduction in serum triacylglycerol (TAG) levels and anti-inflammatory effects [12]. Moreover, n-3 PUFAs play a positive role in the prevention and treatment of metabolic syndrome, including dyslipidemia [13,14] and NAFLD [15]. TAG-type fish oil is most commonly used as a source of n-3 PUFA [16]. In addition, krill oil, which contains phospholipids, has been in the spotlight as an n-3 PUFA-containing oil [17]. The intake of phospholipids has been reported to reduce serum total cholesterol and low-density lipoprotein (LDL) cholesterol levels [18]. It also inhibits intestinal cholesterol absorption [19], atherosclerosis, and liver damage [20] and improves brain function [21,22]. Therefore, as a health-promoting food, krill oil with phospholipids is considered to be better than TAG-type fish oil. In fact, a previous study reported that krill oil intake increases serum high-density lipoprotein (HDL) cholesterol levels more than TAG-type fish oil in humans [23].

*Euphausia pacifica* (*E. pacifica*) (Figure 1A), also called North Pacific krill, is a small, red crustacean similar to shrimp that flourishes in the North Pacific Ocean and is eaten in Japan. *E. pacifica* has large amounts of 8-hydroxyeicosapentaenoic acid (8-HEPE), which has physiological effects. This review discusses the chemical features of *E. pacifica* and the potential benefits of 8-hydroxyeicosapentaenoic acid (8-HEPE) extracted from *E. pacifica* against metabolic syndrome, dyslipidemia, NAFLD, and atherosclerosis in mice under the condition of a high-fat diet (HFD) or Western diet (WD).

## 2. Chemical Features of *E. pacifica*

### 2.1. Lipids of E. pacifica

“Krill” refers to *Euphausiids*, which are widespread in oceans worldwide. *E. pacifica* is a good source of marine n-3 PUFAs, which include EPA and DHA [24]. Both fish oil and krill oil are a source of EPA and DHA; however, the compositions of these oils are different. Fish oil is mostly composed of TAG, and krill oil is mostly composed of TAG and phospholipids. Krill oil also contains the phospholipid form of n-3 PUFA. Because the phospholipid form of n-3 PUFA is incorporated into plasma faster than the TAG form, krill oil can increase the n-3 index at a lower dose in humans [23,25]. Krill oil also contains the antioxidant astaxanthin.

At present, krill oil is generally *E. superba* (Antarctic krill) oil. There are several differences between *E. pacifica* oil and *E. superba* oil. The proportion of phospholipids in *E. pacifica* oil is higher than *E. superba* oil, and the content of oil is higher in *E. superba* than in *E. pacifica*. However, the most conspicuous difference between *E. pacifica* oil and *E. superba* oil is the content of 8-HEPE (Figure 1B), which is higher in *E. pacifica* oil [24]. In contrast, 8-HEPE in fish oil has not been detected. We analyzed the 8-HEPE content in several species, including *Trachurus japonicus*, *Scomber japonicus*, *Haliotis*, *Patinopecten yessoensis*, *Heliocidaris crassispina*, *Pandalus eous*, *Metapenaeopsis barbaraz*, *Marsupenaeus japonicus* and *Balanus rostratus Hoek*, but detected it only in *Pandalus eous, Metapenaeopsis barbaraz*, and *Balanus rostratus Hoek* [24]. Furthermore, the content of 8-HEPE in these crustaceans was less than one-twentieth of that in *E. pacifica*. From our analysis, *E. pacifica* is the best source of 8-HEPE. Furthermore, the 8-HEPE in *E. pacifica* is 8*R*-HEPE, which is metabolized from EPA by 8*R*-lipoxygenase [26,27].

### 2.2. 8-HEPE Extracted from E. pacifica and PPAR Activation

Methanol extract from *E. pacifica* activates the transcription of peroxisome proliferator-activated receptor alpha (PPARα), PPARγ, and PPARδ [28]. In addition, 5-HEPE, 8-HEPE, 9-HEPE, 12-HEPE, and 18-HEPE (hydroxylation products of EPA) obtained from methanol extracts of *E. pacifica* act as PPAR ligands. Two of these products, 8-HEPE and 9-HEPE, enhance the transcription levels of PPARs to a significantly greater extent than 5-HEPE, 12-HEPE, 18-HEPE, EPA, or EPA ethyl ester in NIH-3T3 cells [28]. In fact, 8-HEPE activates the transcription of PPARα, PPARγ, and PPARδ to levels 10, 5, and 3 times greater than EPA, respectively. 8-HEPE also increases the expression levels of genes regulated by PPARs, such as liver fatty-acid-binding protein, enoyl-CoA hydratase/3-hydroxyacyl CoA dehydrogenase, and carnitine palmitoyltransferase, in FaO cells. In contrast to 8-HEPE, EPA at the same concentration has weak or little effect on these gene expression levels and functions, indicating that 8-HEPE is the more potent inducer of physiological effects. As another good source of marine n-3 PUFAs, Antarctic krill oil has been reported to change PPARγ expression in bone diseases, such as osteoarthritis [29] and dexamethasone-induced osteoporosis [30]. Fish oil increases the hepatic mRNA levels of PPARα, liver fatty acid-binding protein, acyl CoA oxidase, cytochrome P450 4a14, and uncoupling protein 2, indicating PPARα activation [31]. It also increases PPARγ protein levels in pancreatic islets [32].

## 3. Potential Benefits of 8-HEPE Extracted from *E. pacifica* against NAFLD and Its Associated Diseases

### 3.1. Effects of 8-HEPE Extracted from E. pacifica on Metabolic Syndrome

Metabolic syndrome is the medical term for a combination of diabetes, hypertension, and obesity and causes dyslipidemia and fatty liver. Moreover, it is associated with greater risk of developing blood vessel diseases, such as coronary heart disease and stroke. The activation of hepatic PPARα could ameliorate body weight gain and improve insulin sensitivity in HFD-fed obese mice [33]. Moreover, adipocyte hypertrophy and the functional disorder of adipose tissue, such as reduced adiponectin secretion, have been reported to be associated with obesity [34]. 8-HEPE (9.5 mg/kg) extracted from *E. pacifica* reduced the amount of visceral fat (Figure 2A) [35], gonadal white adipose tissue, and the size of gonadal adipocyte cells in HFD-fed mice [36]. PPARα activators can increase hepatic fatty acid oxidation and decrease serum TAG levels, which are responsible for adipose cell hypertrophy and hyperplasia, leading to the regulation of obesity. Compared to EPA, 8-HEPE, which is a potent activator of PPARα, might improve metabolic syndrome.

It is well known that adipose tissue regulates energy homeostasis and insulin sensitivity through the secretion of leptin and adiponectin [37]. 8-HEPE increased angiopoietin-like protein 4 expression through PPARδ activation [28,38] more than EPA, leading to the enhancement of glucose uptake [28,39] in mouse myoblasts (C2C12). 8-HEPE (47 mg/kg) also decreased blood glucose levels in WD-fed apoE knock-out (apoE-KO) mice (Figure 2B) [40].

### 3.2. Effects of 8-HEPE Extracted from E. pacifica on Dyslipidemia

Dyslipidemia is defined by abnormal levels of plasma lipoproteins and TAG, and several types are known. Furthermore, dyslipidemia causes several complications, such as NAFLD and atherosclerosis, and its improvement is important for preventing these complications. Fibrates are an important group of drugs used to treat dyslipidemia in clinical practice. They are agonists of PPARα, which plays important roles in the normalization of plasma lipoproteins and TAG levels. GW590735, a PPARα agonist, increased HDL cholesterol and decreased LDL cholesterol, very low-density lipoprotein (VLDL) cholesterol, and TAG in hApoB100/hCETP mice [41]. Plasma TAG levels were significantly decreased in mice fed an HFD with 10 mg/kg 8-HEPE compared with HFD with EPA or HFD alone [36]. In contrast, plasma total cholesterol levels were similar between mice fed an HFD with 10 mg/kg 8-HEPE and HFD only [36]. Interestingly, 8-HEPE (83 mg/kg) suppressed the WD-induced increases in plasma LDL cholesterol in LDL cholesterol receptor knock-out (LDLR-KO) mice [42]. Moreover, it also increased plasma HDL cholesterol levels in WD-fed LDLR-KO mice. These results suggest that a low dose of 8-HEPE is enough to suppress plasma TAG, but a high dose of 8-HEPE might be needed to decrease plasma LDL cholesterol. Although n-3 PUFAs, especially EPA and DHA, play a positive role in the treatment of dyslipidemia [13], these beneficial effects are considered to be mainly due to the ability of n-3 PUFAs to reduce plasma TAG levels [43]. ATP-binding cassette transporter A1 (ABCA1) transports phospholipids and free cholesterol from macrophages to lipid-free apoA-I [44,45], leading to the generation of HDL particles [46,47]. 8-HEPE, but not EPA, increased the gene expression of ABCA1 in murine OxLDL-treated J774.1 macrophages [42]. Rayner et al. [48] showed that the elevated ABCA1 expression increased plasma HDL cholesterol in LDLR-KO mice. Therefore, 8-HEPE may be more effective at increasing circulating HDL cholesterol than EPA.

### 3.3. Effects of 8-HEPE Extracted from E. pacifica on NAFLD

Excessive calorie intake and lack of exercise have contributed to increased obesity and the prevalence of NAFLD in recent years. NAFLD is now the most important cause of chronic liver disease in the absence of excess alcohol consumption. NAFLD includes hepatic steatosis, which may progress to nonalcoholic steatohepatitis (NASH), fibrosis, and cirrhosis. Hepatic steatosis is characterized by the accumulation of TAG lipid droplets in the hepatocyte cytoplasm. Nonesterified fatty acids derived from the plasma are transported into the liver via transport proteins, such as CD36, which causes hepatic TAG accumulation. CD36 is expressed in a variety of cells, such as macrophages, and tissues [49]. Hajri et al. [50] showed that the deletion of nonhepatic CD36 gene expression causes hepatic steatosis and reduces muscle TAG contents in mice. Oil Red O histological staining, a marker of fat accumulation in the liver, was reduced in the liver of LDL-KO mice fed a WD with 83 mg/kg 8-HEPE compared with WD alone [42]. The content of TAG in the liver was also decreased in the WD with 8-HEPE than WD alone in these mice, suggesting that 8-HEPE can improve hepatic steatosis. We showed that 8-HEPE (50 μM) significantly increased CD36 gene expression in OxLDL-treated murine J774.1 macrophages [42]. This effect may improve hepatic steatosis through the relative increase in fatty acid uptake from plasma into the macrophages compared to the liver.

Fatty acids synthesized from glucose in the liver also play a role in the development of hepatic steatosis [51]. Muscle and liver insulin resistance promotes the accumulation of specific lipid metabolites [52]. Insulin resistance also promotes lipogenesis within the liver, leading to the development of hepatic steatosis. IL-6 signaling leads to a STAT3-dependent upregulation of SOCS3, which in turn induces insulin resistance in the liver [53]. Awazawa et al. [54] reported that IL-6 derived from macrophages contributes to the enhancement of hepatic insulin sensitivity through adiponectin. 8-HEPE increased IL-6 gene expression in macrophages. Moreover, 8-HEPE made adipocytes smaller in HFD-induced obese mice, suggesting increased adiponectin release from the adipocytes [37]. Therefore, 8-HEPE may improve hepatic steatosis by improving insulin sensitivity in the liver via highe*r IL-6* gene expression in macrophages and inhibited adipocyte hypertrophy. 8-HEPE decreased serum alanine aminotransferase (ALT) and hepatic TAG in HFD-fed mice [37]. ALT levels in the plasma are used as a marker to indicate hepatic disorders. In fact, Marinho et al. [55] showed that capybara oil decreased plasma ALT levels and improved hepatic steatosis. Moreover, an n-3 PUFA-enriched diet decreased serum ALT levels and damaged areas of the liver, including hepatocyte necrosis, in a Con-A-induced hepatitis mouse model [56]. Our previous studies indicate that 8-HEPE activates PPAR*α* and increases fatty acid oxidation in the liver. In contrast to 8-HEPE, EPA failed to decrease liver TAG levels or plasma ALT or increase the levels of enoyl-CoA hydratase/3-hydroxyacyl CoA dehydrogenase, carnitine palmitoyltransferase, and expression of cytochrome P450 4a14 in the liver of HFD-fed mice [37]. Tanaka et al. [57] showed that PPAR*α* eliminates fatty acids from the liver by increasing the expressions of several genes involved in hepatic fatty acid/triglyceride metabolism. Therefore, 8-HEPE may improve hepatic steatosis by increasing fatty acid oxidation via hepatic PPAR*α* activation.

### 3.4. Effects of 8-HEPE Extracted from E. Pacifica on Atherosclerosis

Several studies have highlighted the association between NAFLD and increased carotid and coronary atherosclerosis [4,58,59]. Palolini et al. [20] demonstrated that Antarctic krill oil inhibits aortic atherosclerosis in WD-fed apoE-KO mice. We elucidated the effects of 8-HEPE extracted from North Pacific krill oil on aortic atherosclerosis using apoE-KO mice. Sudan IV staining demonstrated that 8-HEPE (47 mg/kg) reduced the area of aortic atherosclerosis in WD-fed apoE-KO mice (Figure 3) [40], suggesting that 8-HEPE works as an inhibitor of atherosclerosis. CD36 macrophages participate in atherosclerotic arterial lesion formation by interacting with oxLDL, and CD36 deficiency reduces atherosclerotic lesion formation [60]. Moreover, plasma OxLDL levels were increased in apoE-KO mice [61]. Therefore, 8-HEPE seems to aggravate atherosclerosis by increasing CD36 gene expression in macrophages. However, Moore et al. [62] showed that the loss of CD36 in apoE-KO mice did not alleviate atherosclerotic lesions. Moreover, Zhu et al. [63] showed that the scavenger receptor activity of CD16, which is different from that of CD36, also contributed to the progression of atherosclerosis in apoE-KO mice. Therefore, increased CD36 gene expression in macrophages may not always aggravate atherosclerosis.

## 4. Conclusions

In this review, we explain the potentially beneficial effects of 8-HEPE against metabolic syndrome, dyslipidemia, NAFLD, and atherosclerosis (Figure 4). It is known that fish and krill oils containing n-3 PUFAs have health benefits against NAFLD, dyslipidemia, cardiovascular disease, diabetes, cancer, age-related cognitive decline, and rheumatoid arthritis. However, the beneficial effects of 8-HEPE against cognitive impairment are still unknown. This is an important question for aging societies. In addition, studies regarding 8-HEPE have demonstrated that (1) apoE carrier causes greater increase in EPA-derived 8-HEPE [64] and (2) EPA ethyl esters inhibit HFD-induced fat mass accumulation through EPA-derived 8-HEPE in female mice [65]. These results suggest 8-HEPE plays important roles in human health, even if it is derived from EPA. Further research is needed to investigate the potential benefits of 8-HEPE on human health. Moreover, there are controversial points regarding the effects of n-3 PUFAs on pathological/physiological processes, such as cancer, stroke, diabetes, and brain development, and proper clinical trials of n-3 PUFA-containing therapeutic drugs are lacking because of funding constraints. Therefore, further studies, including clinical investigation, are needed to investigate the beneficial effects of North Pacific krill oil products on human health.

## Figures and Tables

**Figure 1 nutrients-13-03765-f001:**
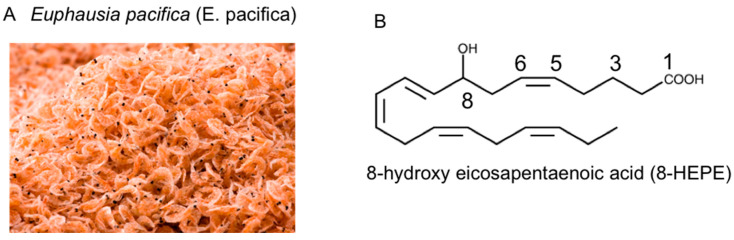
(**A**) Photograph of *Euphausia pacifica* (*E. pacifica*), also called North Pacific krill. (**B**) The structural formula of 8-hydroxyeicosapentaenoic acid (8-HEPE).

**Figure 2 nutrients-13-03765-f002:**
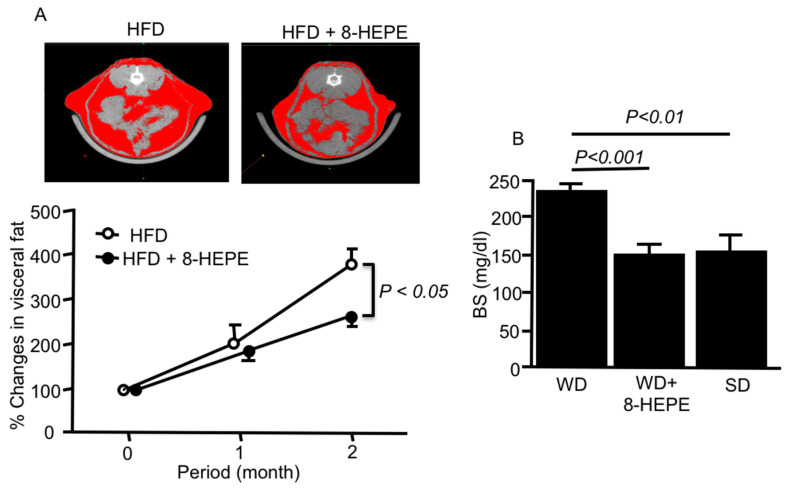
(**A**) Representative computed tomography images of the abdomen (upper) and percentage changes in visceral fat area (lower) in mice fed an HFD only or an HFD with 8-HEPE. Visceral and subcutaneous fat areas are shown in red. Data are the mean ± SE obtained from eight mice for each group. (**B**) Blood sugar (BS) in apoE knock-out (apoE-KO) mice fed a WD only or WD with 8-HEPE. Data are the mean ± SE obtained from 10 mice for each group. An analysis of variance (ANOVA) with Dunnett’s test was used for the statistical analysis. HFD, high fat diet; 8-HEPE, 8-hydroxyeicosapentaenoic acid; WD, western diet.

**Figure 3 nutrients-13-03765-f003:**
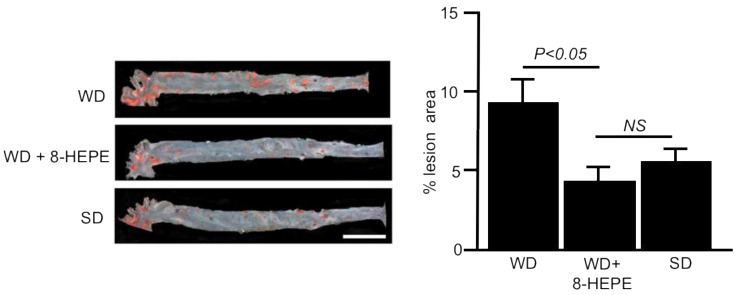
Representative images of the aorta (**left**) stained with Sudan IV and percentage lesion area of atherosclerosis (**right**) in apoE-KO mice fed a Western diet (WD) only or a WD with 8-HEPE (WD + 8-HEPE). Data are the mean ± SE obtained from eight mice for each group. An analysis of variance (ANOVA) with Dunnett’s test was used for the statistical analysis. SD, standard diet; NS, not significant.

**Figure 4 nutrients-13-03765-f004:**
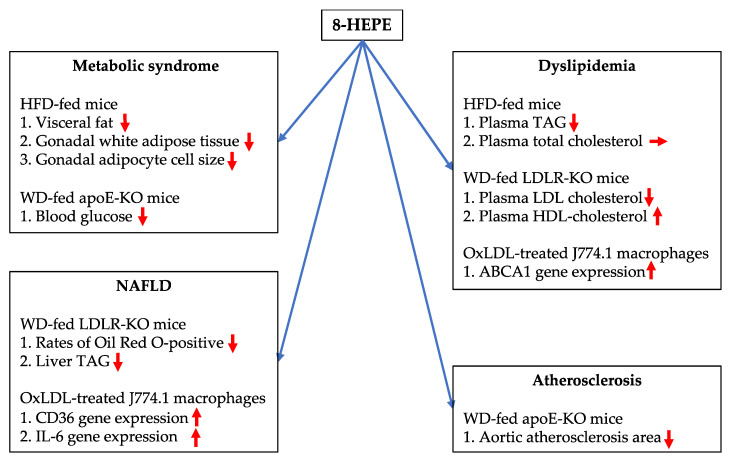
Potential beneficial effects of 8-HEPE against metabolic syndrome, dyslipidemia, NAFLD, and atherosclerosis. HFD, high-fat diet; WD, Western diet; apoE-KO, apolipoprotein E knock out; LDLR-KO, low-density lipoprotein receptor knock out; TAG, triacylglycerol; IL-6, interleukin 6; OxLDL, oxidized LDL; ABCA1, ATP-binding cassette transporter 1.

## Data Availability

Not applicable.

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
