# Peer review of "Euphausia pacifica (North Pacific Krill): Review of Chemical Features and Potential Benefits of 8-HEPE against Metabolic Syndrome, Dyslipidemia, NAFLD, and Atherosclerosis"

_nutrients, 2021, doi:10.3390/nu13113765_

Round 1

Reviewer 1 Report

The manuscript analyses the beneficial effects of Euphausia Pacifica against metabolic syndrome, dyslipidemia, NAFLD, atherosclerosis, and cognitive impairment.

Even if it has been submitted and classified by the authors as a review, the typology of the paper is not clear, since it also includes in some sections (3.8, 8), descriptions, figures and results that look new. In case this is not the case, the authors have to indicate in the Figures the corresponding References and the permission to use the data/images.

Anyway, in the actual structure the manuscript is very difficult to read and several repetitions and inaccuracies are present. For exeample the statement “Moreover, the content of 8-HEPE in E. pacifica oil is more than ten 143 times higher than E. superba oil [24], suggesting….”, is repeated four times in the different paragraphs.

  • Page 1, line 41: the statement “NAFLD, which causes hepatic steatosis” is nor correct, since NAFLD is a hepatic steatosis arising in the absence of alcohol intake.
  • Page 3, line 90. It is not clear why the authors described in a specific paragraph as they prepared some 8-HEPE-containing materials. This description could be included in an Original Article, not in a review. Line, 99: How the different contents of EPA, DHA, 8-HEPE have been chosen?

Moreover, if the results reported in the manuscripts refer mainly to 8-HEPE, also title should underline that the beneficial effects are due to this molecule extracted from Euphausia P.

  • Paragraph 4. The authors evidenced in a previous study that the components of methanol extract from E. Pacifica activate all PPAR isoforms (line 102), but at Page 4, line 136, they report that “Water-soluble extract of E. Pacifica suppressed TAG accumulation in mouse adipocytes by suppressing PPARγ and C/EBPa expression in 3T3-F442A cells”. The authors have to discuss this discrepancy.
  • Paragraph 6. The authors report the ability of 8-HETE in “suppressing” the plasma levels of TG and LDL. This effect can be interesting in animal models, but the total suppression of these circulating lipids is not desirable for health, based on their role in body lipid delivery. A comment/revision needs to be added/made.
  • Paragraph 7. The authors affirm that 8-HETE-induced increased expression of CD36 in macrophages may contribute to improve hepatic steatosis. The consequence of this increased expression on atherosclerosis has to be discussed, since it could contribute to the transformation of macrophages into foam cells, key event in atherosclerotic plaque. In the same paragraph, the comments on the effect of 8-HETE on ALT has to be thorough. Since the decrease of this enzyme in plasma should reflect a prevention of cell death, the authors must verify if in the literature there are studies that have investigated the possible decrease in hepatocyte necrosis.
  • In the Conclusion section, sentences previously reported are repeated and no conclusive comments were made.

The manuscript needs a careful revision to eliminate typos (In Introduction, line 31: The introduction should briefly More than 100 types of liver disease have….”, different indications (n-3 or N-3).

Author Response

Response to Reviewer #1.

The time and effort of the reviewer is greatly appreciated.  The reviewer has raised an important issue that will significantly improve the manuscript. This issue has been addressed below:

Note: In the manuscript text, all changes are shown using “Track Changes” function in MS word.

Reviewer comment: Even if it has been submitted and classified by the authors as a review, the typology of the paper is not clear, since it also includes in some sections (3.8, 8), descriptions, figures and results that look new. In case this is not the case, the authors have to indicate in the Figures the corresponding References and the permission to use the data/images.

Response: As the reviewer pointed out, some of data has not been presented in scientific conferences and published to any publisher. Therefore, we deleted paragraph 3 and the description concerning cognitive impairment in paragraph 8. Moreover, we added 2 references (#35 and 40) concerning 8-HEPE-induced decreases in visceral fat area, blood sugar, and atherosclerotic lesion. The data shown in Figures 2 and 3 have been presented in Society meeting of Cerebral-Cardio-Vascular Anti-Aging 2015 and AHA scientific session 2019, respectively. We think we do not need to get the permission to use the data because there are no published paper regarding them.

Reviewer comment: Anyway, in the actual structure the manuscript is very difficult to read and several repetitions and inaccuracies are present. For example the statement “Moreover, the content of 8-HEPE in E. pacifica oil is more than ten 143 times higher than E. superba oil [24], suggesting….”, is repeated four times in the different paragraphs.

  • Page 1, line 41: the statement “NAFLD, which causes hepatic steatosis” is nor correct, since NAFLD is a hepatic steatosis arising in the absence of alcohol intake.

Response: We corrected it in the revised paper.

  • Page 3, line 90. It is not clear why the authors described in a specific paragraph as they prepared some 8-HEPE-containing materials. This description could be included in an Original Article, not in a review. Line, 99: How the different contents of EPA, DHA, 8-HEPE have been chosen?

Response: As the reviewer pointed out, this description could be included in an Original Article, not in a review. Therefore, we deleted paragraph 3.

Reviewer comment: Moreover, if the results reported in the manuscripts refer mainly to 8-HEPE, also title should underline that the beneficial effects are due to this molecule extracted from Euphausia P.

  • Paragraph 4. The authors evidenced in a previous study that the components of methanol extract from E. Pacifica activate all PPAR isoforms (line 102), but at Page 4, line 136, they report that “Water-soluble extract of E. Pacifica suppressed TAG accumulation in mouse adipocytes by suppressing PPARγ and C/EBPa expression in 3T3-F442A cells”. The authors have to discuss this discrepancy.

Response: We agree the reviewer’s comment. In this review article, we should mainly discuss the effects of 8-HEPE on metabolic syndrome, dyslipidemia, NAFLD, and atherosclerosis. Therefore, we delete the results from the water-soluble extract from E. pacifica in this review article. We also changed Title as follows: “Euphausia pacifica (North Pacific krill): Review of chemical features and potential benefits of 8-HEPE against metabolic syndrome, dyslipidemia, NAFLD, atherosclerosis”.

  • Paragraph 6. The authors report the ability of 8-HEPE in “suppressing” the plasma levels of TG and LDL. This effect can be interesting in animal models, but the total suppression of these circulating lipids is not desirable for health, based on their role in body lipid delivery. A comment/revision needs to be added/made.

Response: Thank you for your comment. It is well known that the increased serum LDL-cholesterol and TG induces dyslipidemia, which causes several complications such as NAFLD and atherosclerosis. In fact, Western diet-fed LDLR-KO mice increased serum LDL-cholesterol levels, leading to the development of atherosclerosis. Therefore, normalization of serum LDL-cholesterol is important for inhibiting atherosclerosis. We added a sentence from line 506 to 508 as follows: “It is known that as dyslipidemia causes several complications such as NAFLD and atherosclerosis, its improvement is important for preventing these complications.”. 

  • Paragraph 7. The authors affirm that 8-HETE-induced increased expression of CD36 in macrophages may contribute to improve hepatic steatosis. The consequence of this increased expression on atherosclerosis has to be discussed, since it could contribute to the transformation of macrophages into foam cells, key event in atherosclerotic plaque. In the same paragraph, the comments on the effect of 8-HETE on ALT has to be thorough. Since the decrease of this enzyme in plasma should reflect a prevention of cell death, the authors must verify if in the literature there are studies that have investigated the possible decrease in hepatocyte necrosis.

Response: Thank you for your comments. Along your suggestion, we discussed association of CD36 with atherosclerosis in page 6, lines 632-639 with 3 references as follows: “Macrophage CD36 participates in atherosclerotic arterial lesion formation through its interaction with oxLDL and CD36 deficiency reduces atherosclerotic lesion formation [63]. Moreover, plasma OxLDL levels were increased in apoE-KO mice [64]. Therefore, 8-HEPE seems to aggravate atherosclerosis through the increased CD36 gene expression in macrophages. However, Moore et al. [65] showed loss of CD36 in apoE-KO mice did not alleviate atherosclerotic lesions. Moreover, Zhu et al. [66] showed that scavenger receptor activity of CD16 different from CD36 also contributed to the progression of atherosclerosis in apoE-KO mice. Therefore, the increased CD36 gene expression in macrophages may not always aggravate atheroscrerosis.”.

We also discussed regarding association of the decerased plasma ALT with the possible decrease in hepatocyte necrosis with 2 references as follows: “In fact, Marinho et al. [56] showed that Capybara oil decreased plasma ALT levels and improved hepatic steatosis. Moreover, n-3 PUFA-enriched diet decreased serum ALT levels and damaged areas of liver including hepatocyte necrosis in Con A-induced hepatitis mouse model [57].”.

  • In the Conclusion section, sentences previously reported are repeated and no conclusive comments were made.

Response: We agree with your comment. We rewrote Conclusion with a figure (Figure 4) as follows: “In this review, we showed 8-HEPE had potential beneficial effects against metabolic syndrome, dyslipidemia, NAFLD, atherosclerosis (Figure 4). It is known that fish and krill oils containing n-3 PUFAs have health benefits against NAFLD, dyslipidemia, cardiovascular disease, diabetes, cancer, age-related cognitive decline, and rheumatoid arthritis. However, the beneficial effects of 8-HEPE against the improvement of cognitive impairment is still unknown. We need to solve the question in recent aging society we encounter. In addition, several studies revealed that there are controversial points regarding the effects of n-3 PUFAs on pathological/physiological processes such as cancer, stroke, diabetes, and brain development. Moreover, proper clinical trials of n-3 PUFA-containing therapeutic drugs are lacking because of funding constraints. Therefore, further studies including clinical investigation are needed to investigate the beneficial effects of pacific krill oil products and 8-HEPE on human health.”.

Reviewer comment: The manuscript needs a careful revision to eliminate typos (In Introduction, line 31: The introduction should briefly More than 100 types of liver disease have….”, different indications (n-3 or N-3).

Response: Thank you for you comments. We carefully revised to eliminate typos in the revised paper.

Reviewer 2 Report

The review by Ishida et al is interesting, pointing to the benefits of pacific krill oil. Yet some points have to be addressed before considering publication.

The introduction, particularly from line 31 to 45, could use a bit more of liaison, and should be rewritten because it reads like a juxtaposition of facts. Same remark goes for lines 78 to 83, because of numerous repetitions. In this paragraph, a bit more of information about 8-HEPE would be welcome though, since it is an important component in krill oil; is krill particularly rich in 8-HEPE, is it found in other food items, how does this compare with krill etc.

There is a lack of transition between the paragraphs, and paragraph 3 seems awkward, with unnecessary precisions.

Generally the text could be alleviated and edited. There are many abbreviations, which makes the text hard to understand. The authors go back and forth between 8-HEPE and omega-3 fatty acids effects, which is confusing. Lines 171 to 177 are hard to follow. So is there an advantage of 8-HEPE on omega 3? What conclusion should be drawn? The reader is not sure to follow what the authors are driving at.

The conclusion is a repetition of facts already mentioned, the authors should end the manuscript by opening on unsolved or urgent questions.

Author Response

Response to Reviewer #2.

The time and effort of the reviewer is greatly appreciated.  The reviewer has raised an important issue that will significantly improve the manuscript. This issue has been addressed below:

Note: In the manuscript text, all changes are shown using “Track Changes” function in MS word.

Reviewer comment: The introduction, particularly from line 31 to 45, could use a bit more of liaison, and should be rewritten because it reads like a juxtaposition of facts. Same remark goes for lines 78 to 83, because of numerous repetitions. In this paragraph, a bit more of information about 8-HEPE would be welcome though, since it is an important component in krill oil; is krill particularly rich in 8-HEPE, is it found in other food items, how does this compare with krill etc.

Response: Along the reviewer’s suggestion, we rewrote the introduction from line 28 to 33 with new references 1-3 as follows: “It is well known that the liver has detoxification, metabolic, and storage functions. When the liver is injured by disease, therefore, those functions are impaired. Obesity-related non-alcoholic fatty liver disease (NAFLD), which is a hepatic steatosis arising in the absence of alcohol intake, impairs the liver's functions. Interestingly, NAFLD is known to participate in the development of obesity-related carcinogenesis [1] and can cause hepatocellular carcinoma [2-3].”. We also added information about 8-HEPE-included food items in paragraph 2 as follows: “In contrast, 8-HEPE in fish oil was not detected. Moreover, when we analyzed 8-HEPE content in Trachurus japonicus, Scomber japonicus, Haliotis, Patinopecten yessoensis, Heliocidaris crassispina, Pandalus eous, Metapenaeopsis barbaraz, Marsupenaeus japonicus and Balanus rostratus Hoek, 8-HEPE was detected in Pandalus eous, Metapenaeopsis barbaraz and Balanus rostratus Hoek [24], however the content of 8-HEPE in these crustaceans was less than one-twentieth than E. pacifica. As far as we analyzed, E. pacifica is a best resource of 8-HEPE.”.

Reviewer comment: There is a lack of transition between the paragraphs, and paragraph 3 seems awkward, with unnecessary precisions.

Response: We added information about each paragraph. We deleted paragraph 3.

Reviewer comment: Generally the text could be alleviated and edited. There are many abbreviations, which makes the text hard to understand. The authors go back and forth between 8-HEPE and omega-3 fatty acids effects, which is confusing. Lines 171 to 177 are hard to follow. So is there an advantage of 8-HEPE on omega 3? What conclusion should be drawn? The reader is not sure to follow what the authors are driving at.

Response: We agree with what the reviewer point out. In fact, the effects of the water-soluble extract from E. pacifica on dyslipidemia differ from that of 8-HEPE. As the reviewer 1 pointed out, in this review article we should mainly discuss the effects of 8-HEPE on metabolic syndrome, dyslipidemia, NAFLD, and atherosclerosis. Therefore, we delete the results from the water-soluble extract from E. pacifica in this review article. We also reduced the number of abbreviations as much as possible.

Reviewer comment: The conclusion is a repetition of facts already mentioned, the authors should end the manuscript by opening on unsolved or urgent questions.

Response: Along the reviewer’s suggestion, we rewrote Conclusion with a figure (Figure 4) as follows: “In this review, we showed 8-HEPE had potential beneficial effects against metabolic syndrome, dyslipidemia, NAFLD, atherosclerosis (Figure 4). It is known that fish and krill oils containing n-3 PUFAs have health benefits against NAFLD, dyslipidemia, cardiovascular disease, diabetes, cancer, age-related cognitive decline, and rheumatoid arthritis. However, the beneficial effects of 8-HEPE against the improvement of cognitive impairment is still unknown. We need to solve the question in recent aging society we encounter. In addition, several studies revealed that there are controversial points regarding the effects of n-3 PUFAs on pathological/physiological processes such as cancer, stroke, diabetes, and brain development. Moreover, proper clinical trials of n-3 PUFA-containing therapeutic drugs are lacking because of funding constraints. Therefore, further studies including clinical investigation are needed to investigate the beneficial effects of pacific krill oil products and 8-HEPE on human health.”.

Round 2

Reviewer 1 Report

The authors rewritten the manuscript following the reviewer's suggestions. In the current form, the manuscritp corresponds more to the criteria of the review.

It is now a short revision of the literature regarding the knowledge on beneficial properties of 8-HETE extracted from Euphausia pacifica. Based on the limited comsumption of this krill, the impact of the manuscript is at the present low, but can contribute to increase the scientific interest on the properties of HETE family.

Author Response

Response to Reviewer #1.

Thank you again for your comment.

Note: In the manuscript text, all changes are shown using “Track Changes” function in MS word.

Reviewer comments: It is now a short revision of the literature regarding the knowledge on beneficial properties of 8-HEPE extracted from Euphausia pacifica. Based on the limited comsumption of this krill, the impact of the manuscript is at the present low, but can contribute to increase the scientific interest on the properties of HEPE family.

Response: Thank you for your important point. However, there are few literatures regarding the knowledge on beneficial properties of 8-HEPE extracted from Euphausia pacifica except our-pulished data.

A few studies regarding 8-HEPE demonstrated that 1) apoE carrier caused a greater increase in EPA-derived 8-HEPE [1] and 2) EPA ethyl esters inhibited high fat diet-induced fat mass accumulation through EPA-derived 8-HEPE in female mice [2]. These results suggest 8-HEPE plays important roles in human health even if EPA is taken. Therefore, further research should be done to investigate potential benefit of 8-HEPE on human health.

  1. Saleh, R.N.M.; West, A.L.; Ostermann, A.I.; Schebb, N.H.; Calder, P.C.; Minihane, A.M. APOE genotype modifies the plasma oxylipin response to omega-3 polyunsaturated fatty acid supplementation in healthy individuals. Front Nutr. 2021, 8, 723813.
  2. Pal, A.; Sun, S.; Armstrong, M.; Manke, J.; Reisdorph, N.; Adams, V.R.; Kennedy, A.; Zu, Y.; Moustaid-Moussa, N.; Carroll, I.; Shaikh, SR. Beneficial effects of eicosapentaenoic acid on the metabolic profile of obese female mice entails upregulation of HEPEs and increased abundance of enteric Akkermansia Muciniphila. Biochim Biophys Acta Mol Cell Biol Lipids. 2021, 59059.

We added information regarding beneficial effects of 8-HEPE on human health described above in Conclusion as follows: “studies regarding 8-HEPE demonstrated that 1) apoE carrier caused a greater increase in EPA-derived 8-HEPE [67] and 2) EPA ethyl esters inhibited high fat diet-induced fat mass accumulation through EPA-derived 8-HEPE in female mice [68]. These results suggest 8-HEPE plays important roles in human health even if EPA is taken. Further research should be done to investigate potential benefit of 8-HEPE on human health.”.

Reviewer 2 Report

The text has been thoroughly transformed and has improved. Yet the introduction should still be restructured and rewritten, and the whole text should be carefully edited for typos (e.g. "chemical futures" in the text) and grammatical mistakes.

Author Response

Response to Reviewer #2.

Thank you again for your comments.

Note: In the manuscript text, all changes are shown using “Track Changes” function in MS word.

Reviewer comment: The text has been thoroughly transformed and has improved. Yet the introduction should still be restructured and rewritten, and the whole text should be carefully edited for typos (e.g. "chemical futures" in the text) and grammatical mistakes.

Response: Along the reviewer’s suggestion, we restructured and rewrote the Introduction. We also edited this revised review article for typos. In fact, English expression of the article was already edited by native English-speaker before original submission. The grammatical mistakes will be edited later by asking native English-speaker again because we have to resubmit it ASAP.